# Protein evidence of unannotated ORFs in *Drosophila* reveals diversity in the evolution and properties of young proteins

**Eric B Zheng, Li Zhao***

Laboratory of Evolutionary Genetics and Genomics, The Rockefeller University, New York, United States

**Abstract** De novo gene origination, where a previously nongenic genomic sequence becomes genic through evolution, is increasingly recognized as an important source of novelty. Many de novo genes have been proposed to be protein-coding, and a few have been experimentally shown to yield protein products. However, the systematic study of de novo proteins has been hampered by doubts regarding their translation without the experimental observation of protein products. Using a systematic, mass-spectrometry-first computational approach, we identify 993 unannotated open reading frames with evidence of translation (utORFs) in *Drosophila melanogaster*. To quantify the similarity of these utORFs across *Drosophila* and infer phylostratigraphic age, we develop a synteny-based protein similarity approach. Combining these results with reference datasets ontissue- and life stage-specific transcription and conservation, we identify different properties amongst these utORFs. Contrary to expectations, the fastest-evolving utORFs are not the youngest evolutionarily. We observed more utORFs in the brain than in the testis. Most of the identified utORFs may be of de novo origin, even accounting for the possibility of false-negative similarity detection. Finally, sequence divergence after an inferred de novo origin event remains substantial, suggesting that de novo proteins turn over frequently. Our results suggest that there is substantial unappreciated diversity in de novo protein evolution: many more may exist than previously appreciated; there may be divergent evolutionary trajectories, and they may be gained and lost frequently. All in all, there may not exist a single characteristic model of de novo protein evolution, but instead, there may be diverse evolutionary trajectories.

**\*For correspondence:**
lzhao@rockefeller.edu

**Competing interest:** The authors declare that no competing interests exist.

## Editor's evaluation

By integrating in silico predictions and mass-spectrometry, this manuscript tackles the problem of annotating the currently nameless stretches of genomic sequence that actually code for proteins. The hundreds of fruit fly protein-coding genes described here offer new inroads for studying some of the very youngest functional elements in genomes, particularly those that have recently emerged from non-coding DNA sequences. This report will be of interest to functional and evolutionary genomics communities.

## Introduction

While the study of the evolution of organismal phenotypes or the fixation of individual mutations has matured over decades, our understanding of how genes evolve is relatively less mature. The dominant mechanism is duplication and divergence (*Ohno, 1970*; *Long et al., 2003*), but the early years of the

genomic era revealed a seeming paradox of species-specific 'orphan genes.' These genes lacked homologs even in closely related species, which made it necessary to invoke nonstandard duplication and divergence mechanisms (*Domazet-Loso and Tautz, 2003*). Advances in both the sequencing of genomes and in comparative genomics soon led to an explosion of work on the origins of orphan genes, generally identified through lineage specificity.

Lineage-specific orphan genes can originate from a variety of mechanisms, ranging from extreme, rapid divergence after duplication (*Chen et al., 2010*; *Domazet-Loso and Tautz, 2003*) to horizontal gene transfer (*Verster et al., 2019*) to de novo gene birth. In the latter mechanism, a previously nongenic genomic sequence acquires genic function, thus yielding a de novo gene that is characterized by its lineage specificity (*Begun et al., 2006*; *Levine et al., 2006*; *McLysaght and Hurst, 2016*; *Schlötterer, 2015*; *Van Oss and Carvunis, 2019*). De novo genes are identified from diverse taxa from fruit flies (*Begun et al., 2006*; *Levine et al., 2006*) to yeast (*Cai et al., 2008*) and even humans (*Knowles and McLysaght, 2009*; *Xie et al., 2012*). While it would seem unlikely for such de novo genes to gain viable function from presumably unoptimized sequence, empirical and population genetics studies have demonstrated that they have important functions (*Cai et al., 2008*; *Lange et al., 2021*; *Levine et al., 2006*; *Reinhardt et al., 2013*; *Xie et al., 2019*; *Zhao et al., 2014*). However, exactly what proportion of lineage-specific genes are contributed through de novo gene birth remains actively studied (*Tautz and Domazet-Lošo, 2011*; *McLysaght and Guerzoni, 2015*; *Vakirlis et al., 2020*).

Despite significant interest, the study of evolutionarily young genes is surprisingly conceptually difficult. While an empirical investigation of biological effect is ideal, designing such experiments for a specific evolutionarily young gene can be challenging due to the likely lack of homology to previously studied genes. Even identifying a given candidate genetic locus as an evolutionarily young gene can be fraught with challenges: evolutionary youth is functionally equivalent to concluding lineage specificity, but methods like phylostratigraphy via BLAST (*Domazet-Loso et al., 2007*), while useful and intuitive, can be vulnerable to false-negative detections, especially for short genes (*Moyers and Zhang, 2015*; *Moyers and Zhang, 2016*). Moreover, while researchers generally agree that a gene should be functional, exactly how this should be defined and assessed can vary substantially (*Van Oss and Carvunis, 2019*). As an example, whether evolutionarily young genes function as RNAs or as proteins is often an area of debate. One of the earliest recognized young genes is an RNA (*Wang et al., 2002*); other studies also had similar conclusions (*Cridland et al., 2022*; *Ruiz-Orera et al., 2015*). In contrast, in several cases, even de novo genes have been empirically shown to yield protein products (*Bungard et al., 2017*; *Cai et al., 2008*; *Knowles and McLysaght, 2009*; *Li et al., 2010*; *Zhang et al., 2019*). While defining a gene as one that can code for a translated protein is convenient and intuitive, evidence for the translation into polypeptides of lineage-specific genes and, in particular, de novo genes as a class is often lacking (*Van Oss and Carvunis, 2019*). As a result, much effort remains needed for systematic approaches to studying evolutionarily young genes and proteins.

Proteomics approaches like shotgun mass spectrometry (MS) can be a powerful tool for the rapid identification of translational evidence for many potential proteins at once and would be a good fit for such a systematic study. However, standard shotgun MS analysis approaches require a priori assumptions about the sequences of the proteins to be detected (*Sinitcyn et al., 2018*). In the case of discovering possible lineage-specific proteins, this a priori requirement can be confounding, as lineage-specific proteins may be systematically under-annotated due to their lack of homology.

Here, we employ a systematic, open reading frame (ORF)-focused MS-first computational approach, to identify unannotated ORFs with evidence of translation (utORFs) in *Drosophila melanogaster*. We further develop a synteny-based protein similarity approach to quantify the comparative genomic similarity of these utORFs across *Drosophila* to estimate their phylostratigraphic age and finally to infer possible de novo origin for the vast majority of these utORFs. Combined with reference datasets on tissue- and life stage-specific transcription and conservation, this data suggests differences amongst these utORFs with respect to their evolutionary and functional properties. Finally, we independently support a subset of these utORFs via additional MS empirical data and ribosome profiling data.

## Results

### A comprehensive database of all possible ORFs in the *D. melanogaster* genome

As a first step toward the systematic discovery of unannotated proteins in *D. melanogaster*, we generated a comprehensive database of all possible ORFs in the *D. melanogaster* genome. This comprehensive database is the result of translating the entire annotated genome in all six frames; we also included the annotated transcriptome to enable discovery of noncanonical translation phenomena such as alternate-frame ORFs (*de Klerk and 't Hoen, 2015*). In this database, each potential ORF is defined as a contiguous span of amino acid (aa)-coding codons; so, these potential ORFs do not necessarily begin with or contain a canonical start codon. This is necessary in order to account for unannotated splicing events, as a six-frame translation of the genome cannot correctly associate the exons of a gene. Since no coding exon within the coding sequence of a gene can contain a stop codon, the set of all codon spans between stop codons is a superset of the set of all exons in the genome. In addition to annotated ORFs (FlyBase r6.15 release), this comprehensive potential ORF database contains 4,582,998 unique potential ORFs that are not otherwise annotated. Of these additional potential ORFs, the median length is 32 aa, and the 95th percentile is 96 aa. Considering solely the 2,597,794 potential ORFs that contain a canonical start codon, the median length following the start codon remains 32 aa, and the 95th percentile is 112 aa. In total, this suggests that the *D. melanogaster* genome has substantial latent potential for generating new protein-coding genes. Of course, the overwhelming majority of these potential ORFs are neither transcribed nor translated and are thus not functional (*Durand et al., 2019*), but considering these will allow us to scan the potential proteomic diversity in *D. melanogaster*.

### Two-round MS search improves discovery of unannotated ORFs without compromising false discovery rate

A comprehensive proteome of *D. melanogaster* was recently published (*Casas-Vila et al., 2017*), including data from 15 stages of the whole lifecycle of the fruit fly (ranging from 0 to 2 hr embryos to adults, including larval and pupal stages), as well as sharper time-resolution data covering 14 stages of embryogenesis (hourly timepoints from 0 to 6 hr; bihourly timepoints from 6 to 20 hr). We analyzed the Casas-Vila et al. data by searching for the complete potential ORF database described above to discover evidence of translation of unannotated ORFs in *D. melanogaster*. Because the number of potential ORFs is several orders of magnitude greater than the number of annotated proteins, the search space is enormously expanded, and thus a traditional one-round search is likely to miss a very large number of identifications at a given false discovery rate (FDR). Accordingly, to improve total sensitivity while maintaining an acceptable FDR, we used two rounds of analysis (Methods). We identify dramatically more novel unannotated protein groups with statistical significance in this two-round approach than in a traditional single-round search, while the number of false-positive identifications is comparable (*Supplementary file 1A*). The substantial enrichment (around 10-fold) of unannotated ORFs relative to either the observed number of false positives or the desired FDR strongly supports the validity of the two-round approach.

In total, after additional filtering (see Methods), our two-round search identified 3123 unannotated unique peptides that underlie 993 unannotated ORFs without homology to annotated proteins (*Figure 1A*; *Figure 1B*; *Figure 1—source data 1*; *Figure 1—source data 2*), which we hereafter refer to as unannotated ORFs with evidence of translation (utORFs). utORFs range in length from 14 aa to over 100 aa, with the longest being 1748 aa (*Figure 1C*). The median length of a utORF is 37 aa, and the mean is 48.6 aa. Among these utORFs, 331 proteins have canonical start codons with 1027 peptides that originate from sequence after the canonical start codon, implying that on average each protein has three supporting peptide-spectrum matches (PSMs). The mean and median lengths of a utORF starting with a canonical start codon are 53.5 aa and 37 aa, respectively, which is very similar to utORFs overall. Unless specifically mentioned, we analyze henceforth the set of 993 utORFs and not only the 331 with canonical start codons.

Compared to the annotated *D. melanogaster* proteome, utORF sequences have similar aa composition (*Figure 1D*; Spearman rank-order correlation of aa frequencies, utORFs to annotated proteome: 0.767, p=7.98E–5). The aa composition for those utORFs with start codons is also highly similar.

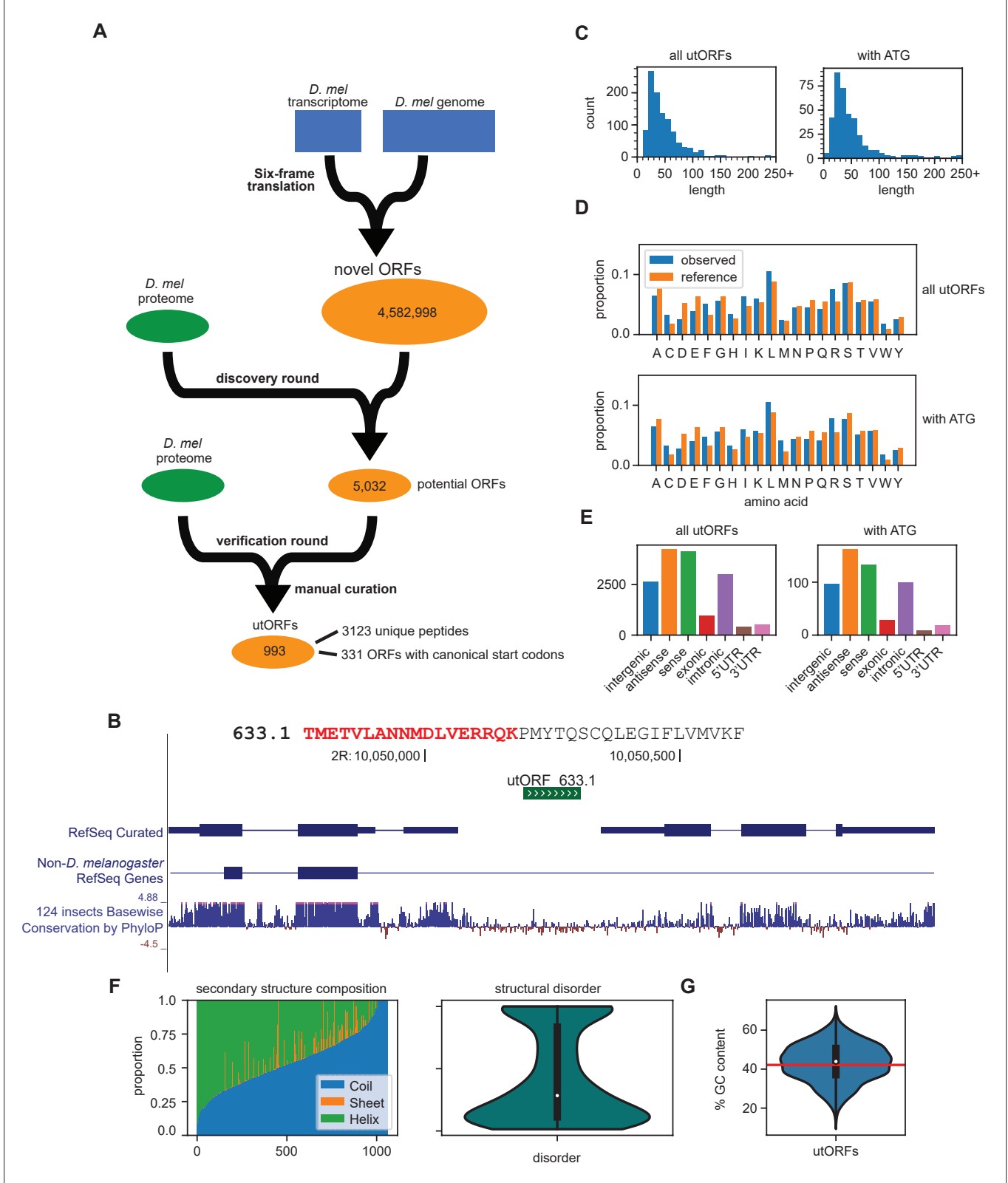

**Figure 1.** Identification of unannotated translated open reading frames (utORFs) and their properties. (**A**) We detected utORFs by searching via a two-round approach through a comprehensive database of potential open reading frames (ORFs) that was generated from a six-frame translation of the transcriptome and genome (see Methods). (**B**) An example utORF. The full sequence of utORF 633.1 is shown. The peptide supporting the inference of this utORF is bolded and colored red. (**C**) Distribution of lengths of utORFs. (**D**) Distribution of amino acids in utORFs (blue) and in the reference *D*.

*Figure 1 continued on next page*

*Figure 1 continued*

*melanogaster* proteome (orange). Standard single-letter abbreviations for amino acids are used. (**E**) Genomic locations of utORFs. Since a utORF can overlap multiple items, e.g., an ORF that overlaps with an annotated intron, exon, and another gene antisense to the first, categories are not exclusive. (**F**) Left: secondary structure composition of utORFs. Approximately half of utORFs have a coiled-coil secondary structure, with the remainder a mixture of beta sheets and alpha helices. Right: predicted structural disorder of utORFs. (**G**) Percent GC content of utORFs and GC content of the X and autosomes of the *D. melanogaster* genome (red line).

The online version of this article includes the following source data for figure 1:

**Source data 1.** All unannotated translated open reading frames (utORFs) sequences.

**Source data 2.** Unannotated translated open reading frame (utORF) supporting peptides.

**Source data 3.** Unannotated translated open reading frame (utORF) locations.

However, some hydrophobic and positively charged aa, such as leucine, arginine, and tryptophan, are overrepresented in utORFs relative to annotated proteins, while negatively charged aa, such as aspartate and glutamate, are underrepresented. Overall, the distribution of positively charged to negatively charged aa is significantly different among utORFs vs. annotated *D. melanogaster* protein sequences (Chi-squared test with Yates's correction, $X^2$=1463.55, df = 1, and p<machine epsilon). In addition, the distribution of hydrophobic aa vs. polar aa is also highly significantly different (same test, $X^2$=44.96, df = 1, and p<1E–10). Since the hydrophobicity and charge of an aa determines its biochemical properties, utORFs may thus differ biochemically from annotated proteins, ranging from differences in secondary structure to propensity and selectivity for protein-protein interactions. These differences in aa composition may be explained by the aa composition of potential ORFs, as the implied aa composition of all potential ORFs is more similar to utORFs than annotated proteins (Spearman rank-order correlation of aa frequencies, potential ORFs to utORFs: 0.970, p=1.71E–12; potential ORFs to annotated proteome: 0.785, p=4.15E–5). This suggests that utORFs have not accumulated many substitutions toward the composition of annotated proteins (e.g. substituting overrepresented positively charged aa for underrepresented negatively charged aa). This could be due to either their evolutionary youth or a lack of selective pressure for such changes.

All 993 utORFs have unique genomic locations (*Figure 1—source data 3*). utORFs reside in a range of genomic locations, including intergenic, intronic, or untranslated regions (UTRs) (*Figure 1E*). About one-third of utORFs are located intergenically; of these, about one-third again are longer than 50 aa and contain a canonical start codon. We propose that these may be single-exon unannotated proteins. A considerable number of utORFs overlap with annotated genes, with an even distribution between sense and antisense directions relative to the annotation. Of the former, several are located either upstream or downstream of the canonical coding sequence (CDS); these may be instances of utORFs or translation read-through, which have been previously described as having important biological functions (*Andrews and Rothnagel, 2014*). Notably, with the exception of a modest reduction in the relative proportion of sense utORFs, the distribution of locations is not different for those utORFs with a canonical start codon. The genomic sequences that make up utORFs on average have greater GC content compared to that of the overall *D. melanogaster* genome (X and autosomes; mean utORF GC content 43.4% vs. X and autosome GC content 42.15%; one-sample two-tailed *t*-test, p<1E–4), but they have wide variety, with some with as much as 62% GC content and others as few as 21% (*Figure 1G*). Intriguingly, the GC content of annotated genes (42.9%) is between that of the genome and utORFs, and utORFs do not have significantly different GC content compared to annotated genes (one-sample two-tailed *t*-test, p>0.05). This suggests either that a selective pressure may act toward increasing the GC content in utORFs or that utORFs may be more likely to originate from regions with higher GC content.

Approximately, half of utORFs (566 of 993; 57.0%) have a majority coiled-coil secondary structure, with the remainder a mixture of beta sheets and alpha helices. Notably, this remains true even when normalizing for length, as the weighted proportion of sequence in a predicted coiled-coil structure is on average about half (*Figure 1F*). Predictions of structural disorder of utORFs suggest that while they are rather disordered, most retain a substantial proportion that is ordered. The median proportion of disordered utORFs is 24.5%. Compared to an empirical null distribution of length-matched potential ORFs without supporting MS evidence, the proportion of coiled-coil of utORFs is statistically

significantly lower (Fisher's combined p-value 0.0182); conversely, the overall proportion of disorder is not significantly different (Fisher's combined p-value 0.682).

## Novel peptides identified are not artifacts caused by polymorphism in conserved genes

It is possible that polymorphisms within a population can yield unique peptides due to aa substitutions, thus resulting in inaccurate attribution of such peptides to utORFs (*Faridi et al., 2018*). However, such polymorphisms can contribute at most a small fraction of the peptides that support the identified utORFs. To quantify this, using a trie data structure for efficient search, we calculated the minimum distance between the peptides that uniquely support utORFs and the expected set of peptides from the reference proteome (see Methods for further details). Of the 3123 tryptic peptides that are uniquely identified from utORFs, only one could possibly result from a single aa substitution of a tryptic peptide from an annotated protein; none could result from a single aa insertion or deletion. A simultaneous double aa substitution could explain an additional five peptides; 24 more peptides could result from a double insertion or deletion. Thus, single or double aa substitutions can explain at most 30 of 3123 unique tryptic peptides (<1%). As a result, it is exceedingly likely that the peptides detected here are the product of utORFs rather than polymorphisms at annotated genes.

## Inferring ages of utORFs

In order to compare the conservation of utORFs across species, it is necessary to find the orthologous sequence with near base-pair-level resolution. Due to the algorithmic compromises necessary for whole-genome searches, BLAST homology searches do not guarantee that they will return optimum results, especially for short sequences that may be highly diverged. Moreover, any homology search that solely relies on searching for a given query sequence necessarily neglects broader genomic context like synteny. Conversely, common genomic utilities like UCSC's *liftOver* are explicitly not recommended for fine-detailed genomic coordinate conversions. To address these issues, we used a pre-existing multiple-sequence alignment (MSA) to guide a maximally optimal local genome alignment (*Figure 2A, B*). This allows the combination of both whole genomic contexts as well as the algorithmic guarantees of the Smith-Waterman alignment.

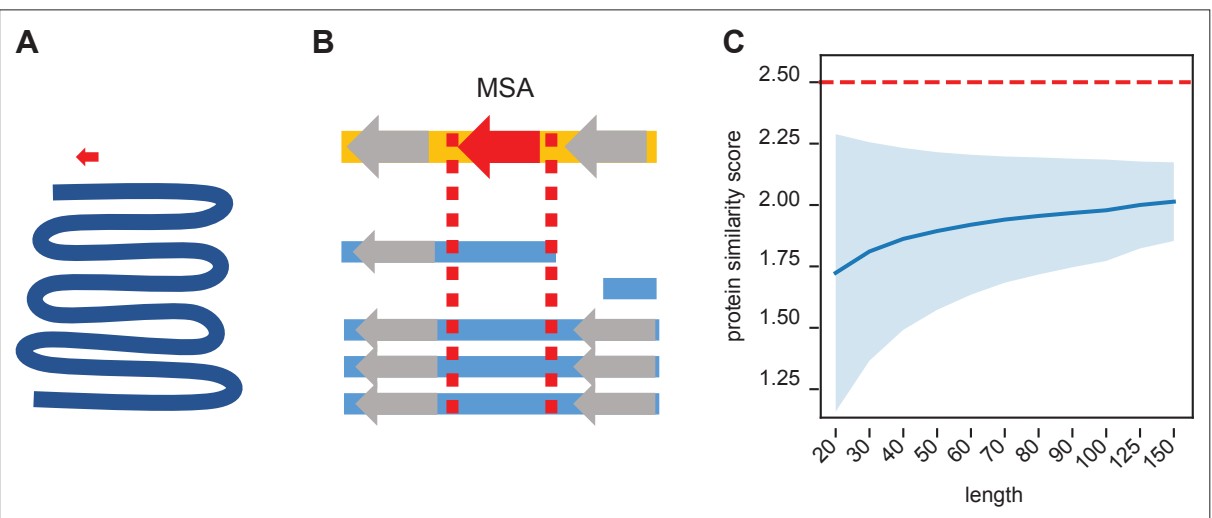

**Figure 2.** Synteny-based orthology detection and protein sequence similarity quantification. (**A**) When performing a simple homology search for a locus of interest (red arrow) across a given genome (blue line), the search space is orders of magnitude larger, requiring heuristic shortcuts to reduce the computational complexity. (**B**) Using a multiple sequence alignment, we can simply find the block that contains the locus of interest and use that to evaluate potential orthologs in other genomes. The search space is approximately a similar size as that of the original locus, so an optimal search is computationally feasible. Syntenic information (i.e. flanking genes, gray arrows) is encoded within the multiple sequence alignment. (**C**) We calculated the pairwise protein similarity score (see Methods) across 10,000 pairs of randomly generated sequences of lengths 20–150 with amino acid frequencies matching that of the annotated *D. melanogaster* proteome. The mean score (blue line) and two SDs (shading) are shown. Our significance threshold of 2.5 lies beyond two SDs from expectation.

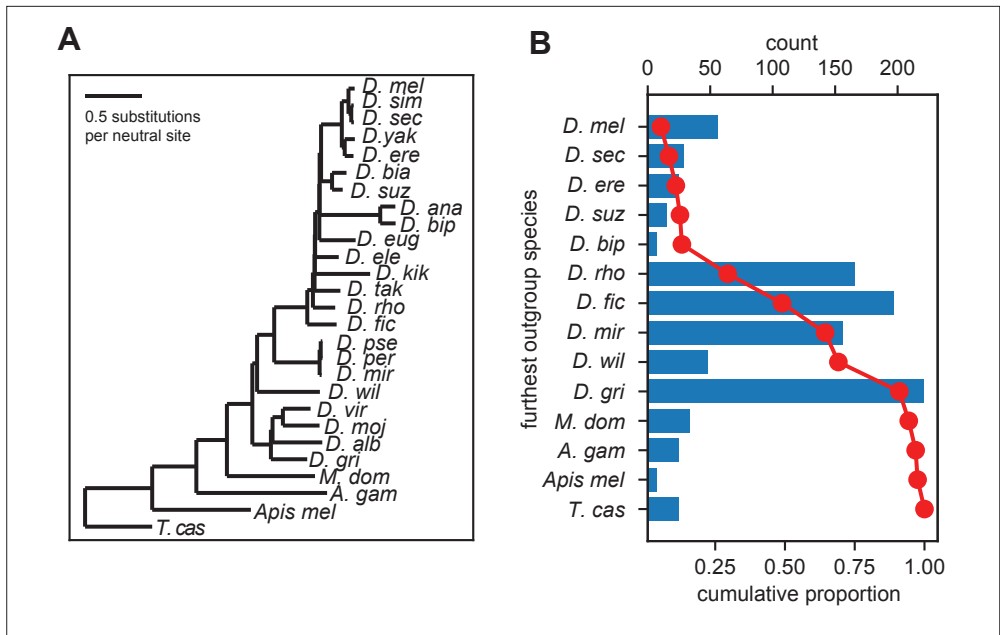

**Figure 3.** Inferred gene ages of the unannotated translated open reading frames (utORFs). (**A**) The reference phylogenetic tree used for these analyses (UCSC 27-way insect alignment). Abbreviations are as follows: *D. mel*: *Drosophila melanogaster*, *D. sim*: *D. simulans*, *D. sec*: *D. sechellia*, *D. yak*: *D. yakuba*, *D. ere*: *D. erecta*, *D. bia*: *D. biarmipes*, *D. suz*: *D. suzukii*, *D. ana*: *D. ananassae*, *D. bip*: *D. bipectinata*, *D. eug*: *D. eugracilis*, *D. ele*: *D. elegans*, *D. kik*: *D. kikkawai*, *D. tak*: *D. takahashii*, *D. rho*: *D. rhopaloa*, *D. fic*: *D. ficusphila*, *D. pse*: *D. pseudoobscura*, *D. per*: *D. persimilis*, *D. mir*: *D. miranda*, *D. wil*: *D. willistoni*, *D. vir*: *D. virilis*, *D. moj*: *D. mojavensis*, *D. alb*: *D. albomicans*, *D. gri*: *D. grimshawi*, *M. dom*: *Musca domestica*, *A. gam*: *Anopheles gambiae*, *Apis mel*: *Apis mellifera*, and *T. cas*: *Tribolium castaneum*. (**B**) The most distantly related species in which a significant ortholog of a utORF exists varies. The red line illustrates the cumulative distribution of loci. For convenience, sister species are grouped together under one species (e.g. *D. virilis* with *D. grimshawi*; *D. eugracilis*, *D. elegans*, *D. kikkawai*, and *D. takahashii* with *D. rhopaloa*; etc.). Abbreviations as in A.

The online version of this article includes the following figure supplement(s) for figure 3:

**Figure supplement 1.** Robustness of gene age inferences with respect to significance threshold.

To estimate a null model of protein similarity scores, we calculated control distributions with varying assumptions. Assuming aa drawn from the distribution of aa in the annotated *D. melanogaster* proteome, 10,000 simulated pairs of randomly generated protein sequences have a mean expected similarity score ranging from 1.7 to 2.0 (*Figure 2C*). We chose a conservative threshold of 2.5 points as a threshold for determining significant sequence similarity, as it is approximately at least two SDs above the mean expectation.

We used the above-described protein similarity score to identify significant orthologs across related *Drosophila* and insect species using the UCSC 27-way MSA (*Figure 3A*). We inferred gene ages for each utORF, where the inferred origin of a utORF corresponded to the last common ancestor between *D. melanogaster* and the most distantly related species where a significant ortholog was identified. We find that utORFs have a wide range of inferred gene ages, ranging from utORFs specific just to *D. melanogaster* to utORFs that may be conserved broadly through ancient insect taxa (*Figure 3B*). The plurality of utORFs likely predate the last common ancestor of *Drosophila* with the most-distant significant ortholog in *Drosophila grimshawi* or any of its sister species; the next largest group predates the last common ancestor of the *Drosophila* species group, with the most-distant significant ortholog in *Drosophila ficusphila* or its sister species. Notably, gene age inferences are not dramatically affected by choice of significance threshold (*Figure 3—figure supplement 1*).

## Latent class analysis reveals subpopulations of utORFs

Latent class analysis (LCA) is a statistical method that supposes that the population under study is composed of a mixture of distinct subpopulations – the 'latent' classes (*Collins and Lanza, 2009*).

**Table 1.** Latent class analysis of all unannotated translated open reading frames (ORFs).

| Class | Interpretation | Estimated percent | Number |
|---|---|---|---|
| 1 | Putatively nonfunctional loci | 4.35% | 41 |
| 2 | *melanogaster*-specific ORFs | 5.71% | 54 |
| 3 | Fast-evolving ORFs | 12.03% | 96 |
| 4 | General unannotated ORFs | 57.61% | 591 |
| 5 | Alternative-frame ORFs | 20.30% | 161 |

Importantly, this is the only major assumption necessary, so the analysis is robust to the shape of distributions or correlations between variables. Moreover, LCA can be used both deductively and inductively; its primary output is a table of probabilities for categorical variables conditional on class membership. Thus, these conditional probabilities can be used deductively to describe differences in the inferred classes, and they can be used inductively to predict class membership given an individual set of observations.

Due to the unbiased strategy of identification, it is likely that the utORFs identified here consist of a mixture of loci resulting from different, distinct evolutionary processes, ranging from intergenic de novo genes to alternative ORFs of conserved annotated genes (*Samandi et al., 2017*). In addition, certain key variables, like inferred gene age and genomic location, are inherently categorical. Rather than simple inter-variable correlations, LCA is a valuable lens for understanding the unannotated loci identified here.

Accordingly, LCA suggests that the population of high-confidence utORFs can be subdivided into five classes (Methods; *Table 1*; *Figure 4—source data 1*) and interpretations assigned based off differences in the various factor probabilities conditional on class membership (*Collins and Lanza, 2009*).

Class 1 loci are notable for minimal transcription (*Figure 4C*) as well as a marked bias of being intergenic or antisense (*Figure 4A*). They also are almost exclusively of intermediate length between 20 and 50 residues (*Figure 4B*). This supports an interpretation as putatively nonfunctional loci. Class 2 loci are remarkable for monophyly (*Figure 4C*), likely due to very recent inferred emergence (*Figure 4D*), suggesting that these may be *melanogaster*-specific loci. Dramatic differences in overall conservation (*Figure 4E*) as well as subtler ones in inferred emergence times (*Figure 4D*) distinguish classes 3, 4, and 5. Class 5 is also remarkable for strong bias toward locations overlapping existing, annotated genes, as well as high conservation, supporting an interpretation as ORFs in alternative reading frames. Finally, class 3 can be distinguished from class 4 through a more recent inferred age of emergence approximately at the last common ancestor of the *melanogaster* subgroup (*Figure 4D*) and through remarkably low conservation that is even more frequent than that of *melanogaster*-specific ORFs. This supports a distinction between class 3 as fast-evolving ORFs and class 4 as general unannotated ORFs.

With inferred classes in hand, it is possible to use the classes for forward inferences by assigning each utORF to a class and comparing the classes. Importantly, while binning continuous variables is necessary for conducting LCA, comparisons between classes can be done on the original continuous variables, removing binning as a potential source of bias, and additional variables not originally used for assigning classes can be used as well. The posterior probabilities of class membership for each utORF suggest generally clean partitioning between classes, as each utORF generally has a high posterior probability of belonging to only a single class (*Figure 4—figure supplement 1*). The exceptions are utORFs with membership in either class 4 (unannotated ORFs) or class 5 (alternative-frame ORFs). However, allowing an additional sixth class leads to overfitting (see Methods). Accordingly, we use the maximum posterior probability per utORF to assign class membership for further analysis.

For certain parameters, differences by inferred class membership recapitulate expected trends. For example, phastCons conservation scores vary by class, with the lowest conservation in fast-evolving and *melanogaster*-specific utORFs (*Figure 5A*). Alternative-frame utORFs are very well conserved due to the additional constraints imposed by their overlapping annotated genes. Despite not being included as a variable in LCA, phyloP conservation scores also vary significantly by class (*Figure 5B*). Negative phyloP scores are consistent with faster evolution; 61.4% (27 of 44) are from fast-evolving

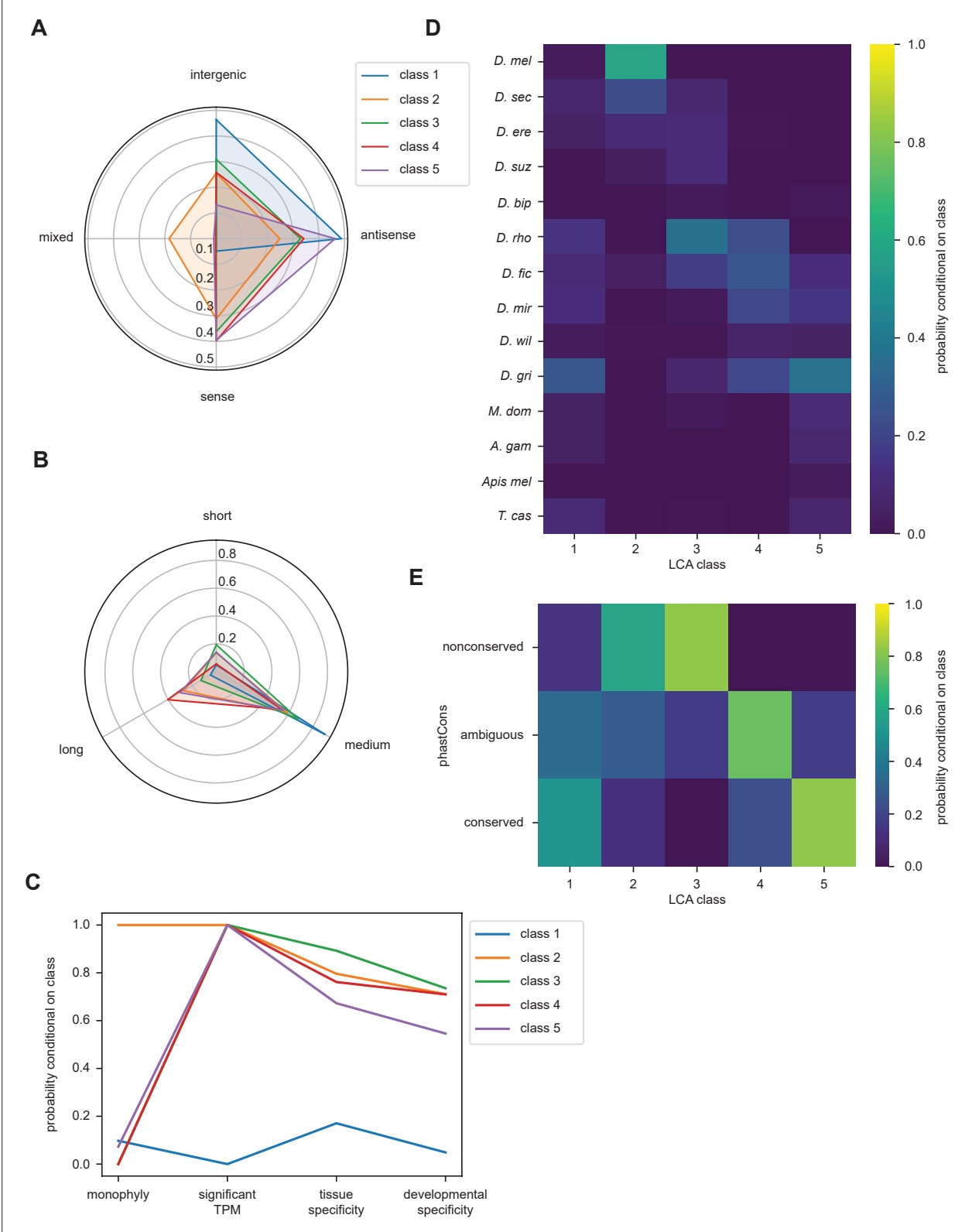

**Figure 4.** Latent class analysis of the unannotated translated open reading frames (utORFs) reveals differences between classes. (**A**) Class 1 is notably distinct for strong bias toward intergenic and antisense locations at the expense of sense locations. Class 2 is notable for being relatively unbiased and for being the only class with appreciable members in a combination of locations. Class 5 is strongly skewed toward antisense and sense locations. (**B**) Class 1 is almost exclusively of intermediate length. Class 4 has the greatest length bias, followed by class 2. Short: fewer than 20 residues; medium:

*Figure 4 continued on next page*

*Figure 4 continued*

from 20 to 49 residues; long: 50 or more residues. (**C**) Class 1 is notably distinct from the others for minimal transcription. Low tissue and developmental specificity may be an effect of minimal transcription. Class 2 is remarkable for being entirely monophyletic. Class 5 has slightly lower tissue and developmental specificities than classes 2–4. significant TPM: maximum per-sample transcripts per million (TPM) > 0.1; tissue specificity: tissue specificity > 0.8; developmental specificity: developmental specificity > 0.8. (**D**) Class 2 is by far the youngest. Class 3 tends to be of an intermediate age, with inferred emergence at around the latent class analysis (LCA) of the *melanogaster* subgroup (*D. rho*). In contrast, class 4 emergence is distributed throughout the LCA of the *melanogaster* subgroup and the *Drosophila* genus. (**E**) Class 2 is notable for overall low conservation. Class 3 is remarkably even less conserved. Classes 4 and 5 are distinguished through differences in intermediate vs. significant conservation. nonconserved: phastCons score < 0.2; ambiguous: phastCons score ≥ 0.2 and < 0.8; conserved: phastCons score ≥ 0.8.

The online version of this article includes the following source data and figure supplement(s) for figure 4:

**Source data 1.** Unannotated translated open reading frame (utORF) inferred latent class analysis (LCA) classes.

**Figure supplement 1.** Posterior probabilities per unannotated translated open reading frame (utORF) of class membership inferred from latent class analysis for all utORFs.

**Figure supplement 2.** Latent class analysis of unannotated translated open reading frames (utORFs) with canonical start sites reveals differences between classes.

**Figure supplement 3.** Posterior probabilities per unannotated translated open reading frame (utORF) of class membership inferred from latent class analysis for utORFs with canonical start sites.

utORFs, and 24.1% (15 of 44) are from *melanogaster*-specific utORFs. Importantly, while phastCons and phyloP scores are significantly correlated, only phyloP scores can differentiate between relative low conservation and accelerated evolution. This recapitulation of faster evolution supports the overall LCA approach. Finally, consistent with the assigned interpretation, *melanogaster*-specific and fast-evolving utORFs are the youngest according to phylostratigraphy (*Figure 5C*). Thus, the fastest-evolving class of utORFs, which is the fast-evolving class characterized by negative phyloP scores, is surprisingly not the youngest class of utORFs, as the distinct *melanogaster*-specific class is younger.

For other parameters, differences between inferred classes reveal surprising trends. Fast-evolving utORFs and putatively nonfunctional utORFs are significantly shorter than all others (one-sided Brunner-Munzel nonparametric test of lengths vs. all other utORFs, p<<0.001 for both). Interestingly, *melanogaster*-specific utORFs are not significantly shorter (same test, p=0.11), despite prior reports that younger genes tend to be shorter. With regards to maximum observed transcription, alternative-frame utORFs tend to be significantly better-transcribed than all other utORFs (one-sided Brunner-Munzel nonparametric test of log-transformed TPMs vs. all other utORFs, p<<0.001). This is likely due to active transcription of the overlapping pre-existing loci. In contrast, fast-evolving utORFs are significantly less well transcribed (same test, p<0.001), while differences in *melanogaster*-specific and utORFs are statistically insignificant. Consistent with these trends, alternative-frame utORFs are less tissue- and developmentally specific (one-sided Brunner-Munzel test of tau across FlyAtlas2 tissues, p<0.001; same test of tau across modENCODE developmental stages, p<0.001), while fast-evolving utORFs are more specifically expressed (tau across FlyAtlas2 tissues, p=0.0039; tau across modEN-CODE developmental stages, p<0.001). Unusually, many of the fast-evolving utORFs (33 of 96, 34.4%) have their highest expression in adult fly tissues in the brain, with the next most common location the testis (10 of 96, 10.4%). Moreover, while transcription of annotated genes is high in both the brain and testis, fast-evolving utORFs are dramatically less-transcribed in the testis than in the brain (75th percentile $\log_{10}$ of TPM in brain, fast-evolving utORFs: 0.172; vs. annotated genes: 1.320; in testis, fast-evolving utORFs: –0.662; vs. annotated genes 1.177; delta $\log_{10}$ of TPM in testis vs. in brain, fast-evolving utORFs vs. annotated genes: one-sided Mann-Whitney U test p=7.5E−6; *Figure 5—figure supplement 1*). Indeed, with the exception of alternate-frame utORFs, most utORFs are not transcribed in the testis but are transcribed in the brain, while most annotated genes are highly transcribed in both (*Figure 5—figure supplement 1*). This is somewhat intriguing as previous work suggests that evolutionarily young genes – particularly de novo genes – emerge in the testis.

When considering only utORFs with canonical start codons, the results are generally qualitatively similar (*Figure 4—figure supplement 2*, *Figure 5—figure supplement 2*). Five latent classes remain the best fit, as allowing a sixth class has unfavorable Akaike information criterion (AIC) and Bayesian information criterion (BIC) effects. Class assignments can proceed with similar logic; however, utORFs with start codons are longer overall (*Figure 4—figure supplement 2B*), so some class interpretations change somewhat. Class 3 appears biased toward intergenic locations and longer length, so we

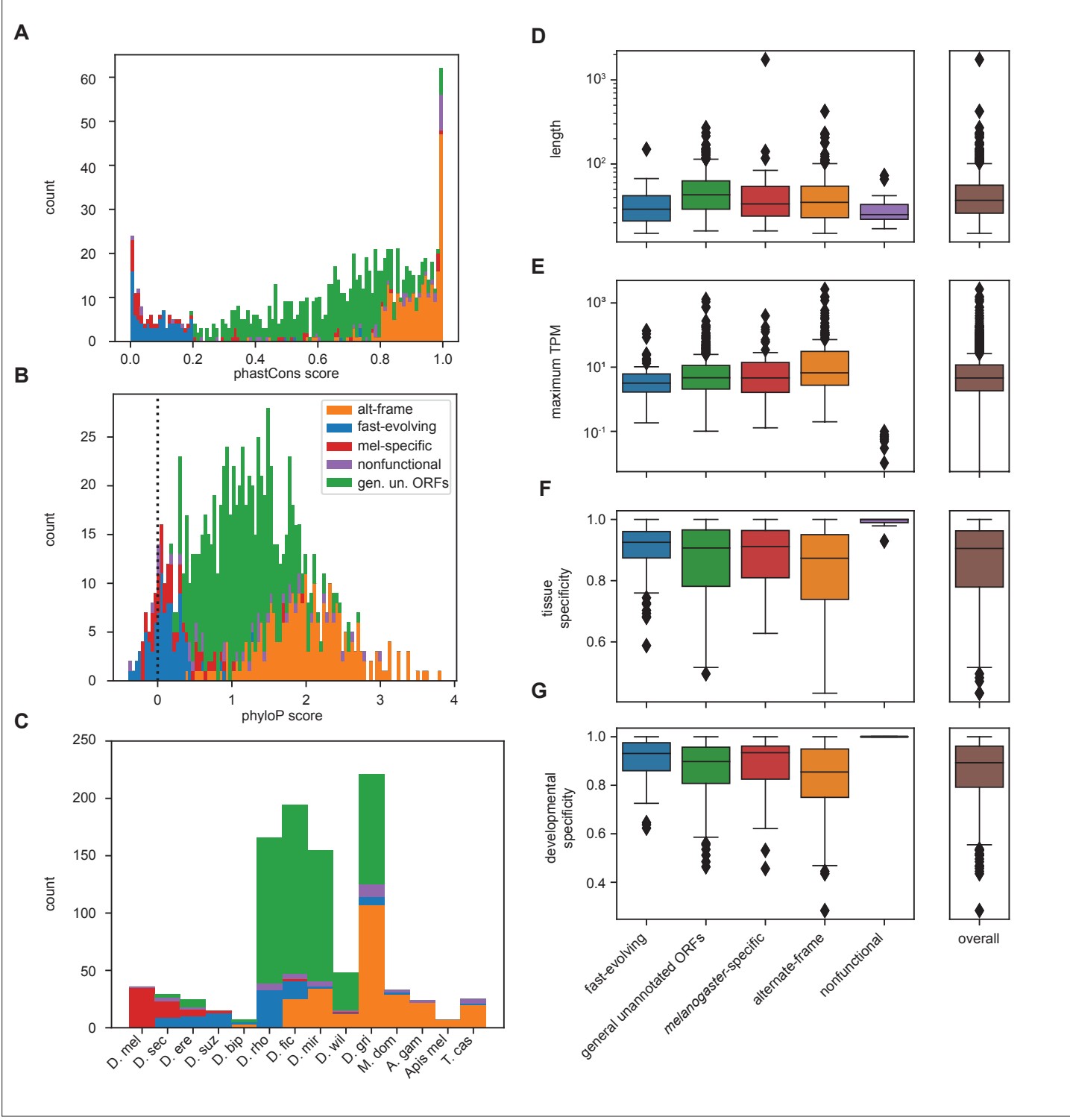

**Figure 5.** Differences between inferred classes recapitulate expected trends in age and conservation and reveal surprising trends in lengths and expression. (**A**) As expected, phastCons conservation scores vary by class. Scores near 0 indicate low conservation, while scores near 1 indicate high conservation. Note that fast-evolving and *melanogaster*-specific loci have dramatically lower conservation, whereas alternate-frame loci are very well conserved. (**B**) Despite not being included as a variable in latent class analysis (LCA), phyloP conservation scores also vary by class. Scores below 0 indicate potential fast evolution (acceleration), while scores above 0 indicate conservation. Note that the lowest scores predominantly occur in fast-evolving and *melanogaster*-specific loci and that alternate-frame loci remain the best conserved. (**C**) Distribution of phylostratigraphically inferred gene ages by inferred latent class. (**D**) Class variations in length are generally minimal, with the exception of general unannotated open reading frames (ORFs)

*Figure 5 continued on next page*

*Figure 5 continued*

being slightly longer than average, and fast-evolving and nonfunctional loci being shorter than average. (**E**) Maximum observed transcription across all FlyAtlas2 and modENCODE samples varies by class. (**F**) Tissue specificity (tau) calculated from FlyAtlas2 data shows that nonfunctional and fast-evolving loci are the most specific, with alternate-frame loci the least. (**G**) Developmental specificity (tau) calculated from modENCODE data similarly shows that nonfunctional and fast-evolving loci are the most specific, with alternate-frame loci the least.

The online version of this article includes the following figure supplement(s) for figure 5:

**Figure supplement 1.** Transcription of unannotated translated open reading frames (utORFs) in selected tissues.

**Figure supplement 2.** Differences between inferred classes for unannotated translated open reading frames (utORFs) with canonical start sites recapitulate expected trends.

assign it an interpretation of 'intergenic ORFs' (*Supplementary file 1B*). Partitioning between classes remains good (*Figure 4—figure supplement 3*). Turning once again toward forward inferences, we recapitulate the patterns seen for the conservation of *melanogaster*-specific and alternative-frame ORFs. While differences in length are apparent between classes (*Figure 5—figure supplement 2D*), they are not statistically significant; the *melanogaster*-specific class still is not significantly shorter (one-sided Brunner-Munzel nonparametric test of lengths vs. all other utORFs: intergenic ORFs, longer, p=0.11; alternate-frame, shorter, p=0.15; *melanogaster*-specific, shorter, p=0.36; putatively nonfunctional, shorter, p=0.20). Finally, utORFs with canonical start sites remain less transcribed in the testis than the brain (delta log base 10 of TPM in testis vs. in brain, utORFs with canonical start sites vs. annotated genes: one-sided Mann-Whitney U test p=7.3E−21). Noncanonical start initiation is increasingly recognized, especially for short ORFs; evidence for translation initiation at sites other than AUG has come from both ribosome profiling and MS approaches from a variety of model species (*Chen et al., 2020*; *Ingolia et al., 2011*; *Ma et al., 2014*; *Wu et al., 2020*). Since we recapitulated similar results for utORFs with canonical start sites as for all utORFs, we continued with all 993 utORFs.

## Many utORFs have evidence consistent with a de novo origin

A genetic locus emerging through de novo gene birth from nongenic sequence would be expected to be detected in a lineage-specific manner with detectable genomic homology despite evidence inconsistent with functional conservation. That is, a utORF may be of de novo origin if there exists at least one species where an orthologous putative ORF is not significantly similar by our protein similarity score (*Figure 2C*) despite genomic homology. Ideally, the strongest inferences of de novo origin would rely on multiple outgroup species. This protects against an erroneous de novo inference due to a homology detection failure rather than true functional changes, which has been extensively described as a potential source of erroneous identifications (*Moyers and Zhang, 2015*; *Moyers and Zhang, 2016*; *Moyers and Zhang, 2018*). Many of our utORFs have such stringent evidence consistent with a de novo origin (*Figure 6A*), with at least two supporting outgroups originating from different phyla. Between 40 and 75% of identified loci across all inferred classes may be de novo, with the greatest proportion in the fast-evolving, *melanogaster*-specific, and unannotated ORFs. These differences between classes are statistically significant (Pearson's Chi-squared test, 4 degrees of freedom, $X^2$=31.941, and p=1.97E−6). For all utORFs with potential de novo origin, 394 (67.8%) have three or more such supporting outgroup species. The number of such supporting outgroup species appears to be more affected by the availability of such than by genomic divergence, as the youngest classes (*melanogaster*-specific and fast-evolving) have more supporting outgroup species than the comparatively older alternative-frame class (*Figure 6B*).

Finally, we considered the frequency by which utORFs of potential de novo origin are lost. Since we consider the protein sequence to be equivalent to the function, we use similar logic as for finding support for de novo origin, but looking only at species descended from the inferred potential de novo origin event. We find that losses of utORFs of potential de novo origin are extremely common; for all utORFs with potential de novo origin, 470 (80.9%) have at least one potential inferred loss (*Figure 6C*).

## Observed retention times of many peptides underlying utORFs are similar to those of peptides underlying annotated proteins

One possible weakness of the methodology used in this study is that PSMs are prone to false identifications (*Nesvizhskii and Aebersold, 2005*). One method that has been proposed to evaluate PSMs

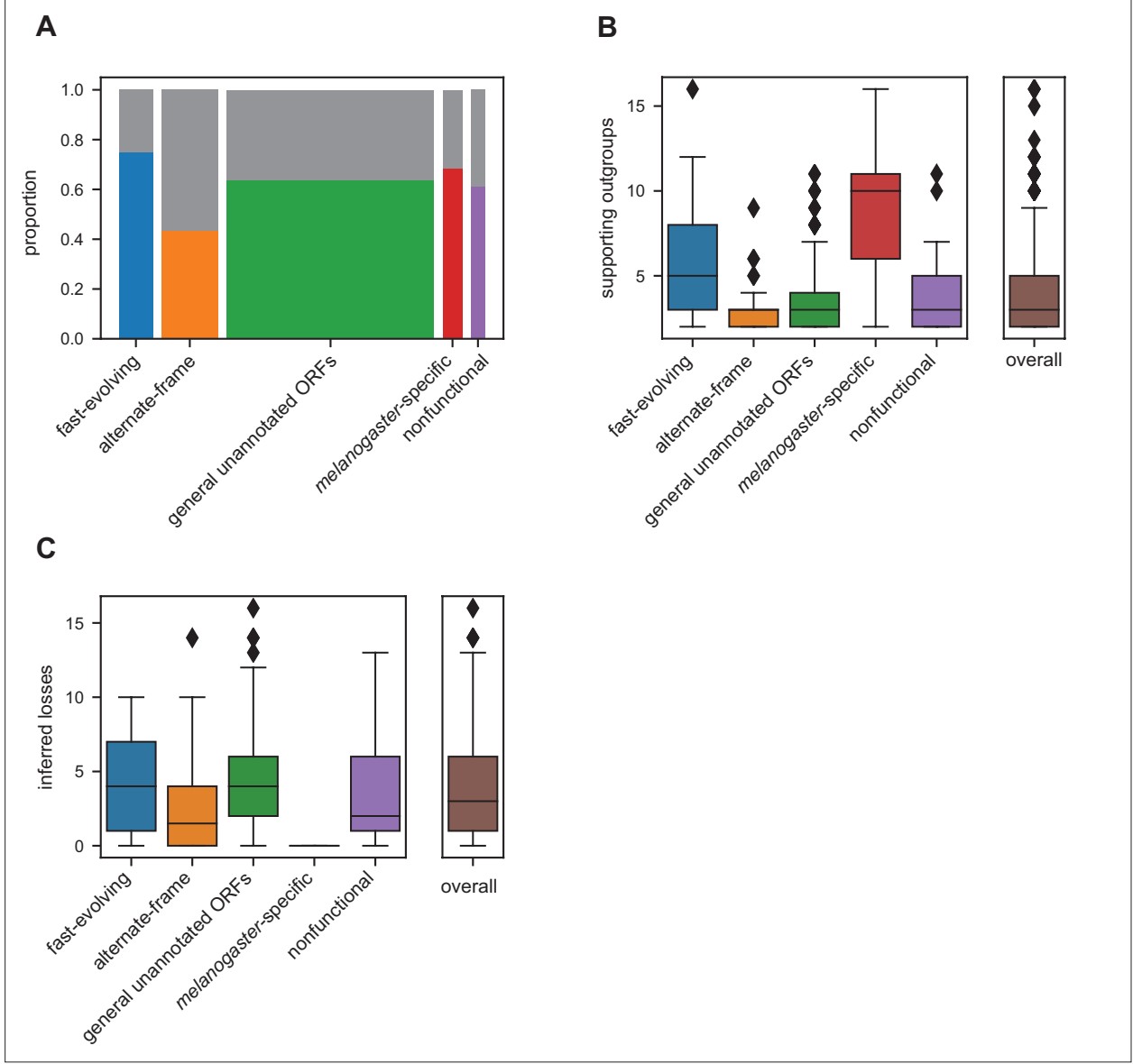

**Figure 6.** Many unannotated translated open reading frames (utORFs) have evidence consistent with a de novo origin. (**A**) Proportion of utORFs by inferred class with genomic conservation consistent with de novo origin. Box widths correlate with size of class (**Table 1**). (**B**) Number of supporting outgroups by inferred class for loci with potential de novo origin. (**C**) Number of losses inferred after potential de novo origin.

is to use the retention time of identified peptides on the liquid chromatography system upstream of the mass spectrometer; retention time should vary as a function of the peptide's sequence. While calculating retention time of a given peptide a priori remains an unsolved problem, substantial progress has been made on predicting retention times of sets of peptides a posteriori. To evaluate the validity of PSMs of utORFs, we used AutoRT (**Wen et al., 2020**) to predict retention times of peptides supporting annotated FlyBase proteins as well as utORFs. While some peptides supporting utORFs have drastically larger differences from the predicted retention times (**Figure 7—figure supplement 1**), a sizeable number have observed retention times that are similar to peptides from annotated FlyBase proteins. Specifically, 385 of 3123 peptides (12.3%) supporting utORFs have at least one PSM with a delta retention time that is smaller than the 95th percentile of the delta retention times of all PSMs that support annotated FlyBase proteins; within this subset, the difference between predicted and observed retention times is modest (**Figure 7A**). These peptides are sufficient to support 270 utORFs, where *melanogaster*-specific ORFs are slightly more likely and nonfunctional ORFs somewhat

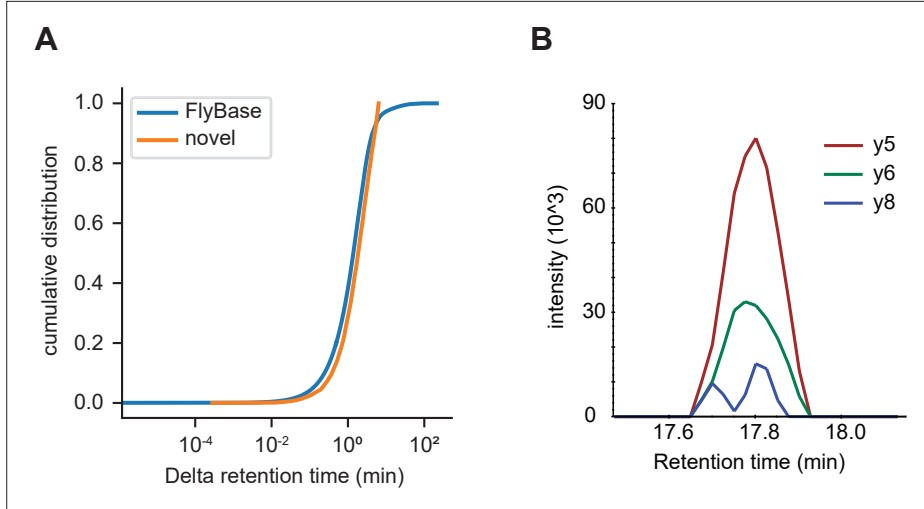

**Figure 7.** Independent validation of unannotated translated open reading frame (utORF) identification. (**A**) Cumulative distribution of differences between observed and predicted retention times for peptide-spectrum matches (PSMs) of peptides supporting annotated FlyBase proteins (orange) and PSMs of peptides supporting utORFs that are no worse than the 95th percentile of annotated FlyBase PSMs (blue). (**B**) Transitions detected via parallel reaction monitoring (PRM) of the peptide GPGAAISQR from protein extracts of *D. melanogaster* embryos.

The online version of this article includes the following source data and figure supplement(s) for figure 7:

**Source data 1.** Mascot search results from embryo mass spectrometry (MS) data.

**Source data 2.** Rank of potentially biologically significant targets.

**Source data 3.** Dataset subset mappings.

**Figure supplement 1.** Cumulative distribution of differences between observed and predicted retention times for every peptide-spectrum match of peptides supporting all annotated FlyBase proteins (blue) and all unannotated translated open reading frames (utORFs) (orange).

**Figure supplement 2.** Proportion of unannotated translated open reading frames (utORFs) by inferred class with supporting evidence from ribosome profiling.

less likely to be supported (*Supplementary file 1C*); however, this observation is not statistically significant (Chi-squared test with 4 degrees of freedom; $X^2$=4.45; p>0.05).

## utORFs can be independently validated

To further validate that the discovered utORFs are not artifactual, we collected our own data to develop independent empirical support for these utORFs. We collected targeted MS data from mixed 0 to 24 hr *D. melanogaster* embryos (RAL517 strain), targeting a manually curated set of 15 target utORFs chosen both for suitability for targeted MS (e.g. few missed cleavages; no post-translational modifications) as well as expression in embryo from modENCODE data. Of these 15 targeted utORFs, we were able to detect supporting evidence for one through the observation of three transitions (*Figure 7B*). We also collected shotgun MS data from mixed 0 to 24 hr embryos; we were able to detect supporting evidence for additional utORFs from the shotgun MS data at a FDR of 1% (*Figure 7—source data 1*); this shotgun dataset was fairly shallow, as we identified 1963 annotated proteins compared to the approximately 8000 protein groups reported by *Casas-Vila et al., 2017*. Given the low sensitivity of MS, having evidence for utORFs from independent datasets likely supports a substantial translation level for at least a subset of the utORFs. Furthermore, we reanalyzed published ribosome profiling data (*Patraquim et al., 2020*; *Zhang et al., 2018*) and identified supporting evidence from an additional 243 utORFs. Of these, alternate-frame ORFs are unsurprisingly the most frequently supported, perhaps due to translation of the canonical ORF, but the other classes are well represented with the exception of the putatively nonfunctional utORFs (*Figure 7—figure supplement 2*). The biases across classes are statistically significant (Pearson's Chi-squared test, 4 degrees of freedom, $X^2$=41.569, and p<<0.001), and they are consistent with our proposed interpretation of the classes. Finally, we specifically looked for evidence of framing in ribosome profiling data by implementing the

binomial statistical test used for the first report of framing in *Drosophila* (*Patraquim et al., 2020*). Of 81 utORFs with more than 2 unique footprints, 17 utORFs (21.0%) have statistically significant biases toward the observed frame in at least one stage of embryogenesis, supporting the notion of active translation (alpha 0.05; 11 utORFs at alpha 0.001).

Together, this evidence supports our identification of the pool of utORFs overall. We accordingly integrated all data that we analyzed with respect to additional lines of evidence supporting our utORFs and factors of interest in order to identify utORFs of highest priority for downstream functional analyses; we used a combination of high transcription from RNAseq atlases, supporting validation from either our MS experiments or framed ribosome profiling reads and conservation metrics. By calculating ranks across each of these factors and averaging ranks per utORF, we generated a priority list for our utORFs (*Figure 7—source data 2*). Of the top 50 utORFs, 24 are alternate-frame ORFs, 14 unannotated ORFs, 7 melanogaster-specific ORFs, and 4 fast-evolving ORFs. This is similar to the overall proportion of class assignments, except alternate-frame ORFs are somewhat over-represented due a higher incidence of framed ribosome profiling reads. These utORFs may be the most promising candidates for future experimental work.

## Discussion
### Identification of hundreds of potential protein products of de novo origin

In this work, we identify almost 1000 possible utORFs from a systematic, ORF-focused MS-first computational approach. These utORFs are unlikely to be annotation failures due to the lack of sequence homology to annotated proteins. Instead, like many similar 'orphan genes,' these may be of de novo origin: the vast majority have a syntenic sequence conservation pattern that is consistent with de novo origin (*Figure 6AB*). The total number of identifications is larger than previous studies of de novo genes within a relatively short timescale in *Drosophila* (*Begun et al., 2006*; *Chen et al., 2010*; *Zhao et al., 2014*) and is consistent with the identification of thousands of possible de novo polypeptides from systematic studies in other biological systems (*Ruiz-Orera et al., 2018*; *Stein et al., 2018*; *Durand et al., 2019*). Moreover, while evidence of translation of proposed de novo genes is generally lacking (*McLysaght and Hurst, 2016*), in this case, the method of identification allows a stronger inference of translation.

Calling de novo origin for a given gene is challenging, as the call is almost wholly dependent on negative results in the outgroup lineages. For example, in small phylogenies with a single outgroup, it is possible for a single false-negative call to lead to an incorrect inference of de novo origin. Recent studies have demonstrated that false negatives may be alarmingly likely, especially when using simple BLAST-based phylostratigraphy (*Moyers and Zhang, 2015*; *Moyers and Zhang, 2016*). Here, our method of calling de novo origin employs proposed best practices of incorporating syntenic information (*Arendsee et al., 2019*; *Vakirlis and McLysaght, 2019*) and is robust to individual sequence homology calls, improving our confidence in our inferences. Importantly, our inferences of gene age and potential de novo origin do not rely at all on the absence of sequence from the MSA, which would be analogous to the lack of a significant BLAST hit. Instead, our inferences of de novo origin rely on protein similarity scores below our threshold of significance (*Figure 6B*), which ensures that these inferences are not simply due to extensive genomic divergence beyond the limit of detection. If anything, our formulation of the protein similarity score may be vulnerable to calling spurious similarity due to shared genomic ancestry between closely related species. However, such errors would cause us to infer systematically older de novo origin dates for each gene; given that we still conclude that many utORFs are of potential de novo origin, the impact of such possible errors is low.

Other mechanisms beyond de novo origin could potentially produce 'orphan'/lineage-specific genes. For example, horizontal gene transfer from microorganisms like viruses has been reported in *Drosophila* (*Verster et al., 2019*), and much attention has been focused on sequence divergence (e.g. after pseudogenization) as an alternative explanation for lineage-specific genes (*Moyers and Zhang, 2015*; *Weisman et al., 2020*). When, as is often the case for our utORFs, there are many supporting outgroups, and multiple pseudogenization events are necessary by parsimony to produce the observed pattern; while this seems unlikely, we cannot exclude the possibility of repeated substantial divergence. However, work explicitly focusing on the possibility that sequence divergence

is responsible for lineage-specific genes recently concluded that extensive divergence was not likely the dominant mechanism (*Vakirlis et al., 2020*).

Our confidence in our identifications of utORFs is supported by our confirmatory results from MS and published ribosome profiling, including the observation of statistically significant framing from ribosome-profiling data. While the absolute number of recapitulated utORF detections is relatively low, MS has notoriously low power, even in well-studied proteomes like humans (*Baker et al., 2017*). Of note, our validation data was collected from embryos of mixed ages ranging from 0 to 24 hr, while the original identification data was collected from multiple narrowly staged embryonic development timepoints. Thus, it is not particularly surprising that we were unable to recapitulate most utORFs in a relatively heterogenous sample, as we expect utORFs to be of low abundance and potentially only expressed under specific time- and tissue-limited conditions. Unlike sequencing approaches, the empirical readout of MS is not directly mappable to information (i.e. the underlying nucleotide sequence), so false identifications of PSMs are a notorious problem in the field (*Nesvizhskii and Aebersold, 2005*). However, the peptides underlying our identification of utORFs are not artifacts caused by polymorphism in conserved genes. Moreover, our results suggest that the reverse-decoy approach for controlling the number of false-positive identifications remains valid for our two-round method; the power for the identification of annotated *Drosophila* genes is nearly the same between a standard MS approach and our two-round approach (*Supplementary file 1A*). Further functional investigation of utORF products would be a potential extension of this work (*Figure 7—source data 2*).

## Class inference reveals heterogenous evolutionary dynamics

In contrast to previous work, our approach of applying LCA allows additional refinement beyond that achievable by simple per-variable analyses. For example, previous studies have described an association between gene age and length: namely, that younger genes tend to be shorter (*Carvunis et al., 2012*; *Neme and Tautz, 2013*; *Wilson et al., 2017*). Here, we find that the youngest class of utORFs (the *melanogaster*-specific class) is not significantly shorter; instead, the fast-evolving and the nonfunctional classes are the shortest overall (*Figure 5C*). Importantly, while length is an input into the LCA, it is neither the most obvious differentiator between the classes nor is it the primary basis upon which we interpreted the classes. Since the fast-evolving class is still quite young, it is possible that previous reports of an association between gene age and length are related to a conflation of fast evolution with gene age. Importantly, our analysis is not able to evaluate whether there is a direction in the historical evolutionary trajectory of the length of a given utORF. Thus, we cannot distinguish definitively between possible causes and effects for a relationship between fast evolution and length. For example, it is possible that shorter utORFs are faster evolving (e.g. due to fewer epistatic constraints) or that faster evolving ORFs tend toward a shorter length (e.g. by avoiding the processes that contribute to increasing gene length over time). This question merits further examination in future.

Previous studies, including our own, have suggested that evolutionarily young genes tend to be expressed in the testis (*Levine et al., 2006*; *Begun et al., 2007*; *Soumillon et al., 2013*; *Zhao et al., 2014*; *Luis Villanueva-Cañas et al., 2017*; *Witt et al., 2019*), though other tissues, like the brain, can also give rise to evolutionarily young genes (*Chen et al., 2012*). In contrast, the utORFs identified here are expressed more in the brain than in the testis (*Figure 5—figure supplement 1*) in each inferred class. Moreover, while most annotated genes are highly expressed in the testis, most utORFs are not expressed at all in the testis, yet they are expressed in the brain (*Figure 5—figure supplement 1*). Considering that most previously reported de novo genes also tend to be expressed in the testis (*Levine et al., 2006*; *Begun et al., 2007*; *Zhao et al., 2014*; *Luis Villanueva-Cañas et al., 2017*; *Witt et al., 2019*), our results may suggest that the proportion of protein-coding de novo genes might be higher in the brain than in the testis or that our MS-first approach allows the detection of otherwise-missed protein-coding de novo genes. Furthermore, these utORFs may have biologically significant effects in the brain. For example, the maximum observed expression of many of the relatively short, fast-evolving ORFs is in the brain, suggesting that some could act as neuropeptides.

Despite the success of our LCA approach in elucidating distinct evolutionary trajectories, half of our identified utORFs of interest remain consolidated in a single class. However, additional classes

cannot be safely supported due to overfitting. Additional approaches may be necessary in order to further characterize these utORFs.

## Turnover in utORFs may be frequent

While de novo gene birth has been increasingly well studied over recent years, de novo gene death has been examined less closely. It is obvious that de novo gene death must be relatively frequent, as otherwise the net size of proteomes would be unrealistically large relative to the annotation. Consistent with this intuition, we observe that most de novo genes diverge by enough to be indistinguishable from random sequences in at least part of the clade descended from the last common ancestor after the de novo gene birth event (*Figure 6C*). Since we consider conservation of the protein-coding sequence to be approximately equivalent to the function of the locus, we consider such sequence divergence as evidence of gene death. Other studies of young genes, including in *Drosophila*, have similarly concluded that gene death among young genes is frequent (*Palmieri et al., 2014*; *Schmitz et al., 2018*; *Stein et al., 2018*).

An implication of most models of de novo gene origins is that de novo gene birth is rare at the individual gene level and thus occurs only once in evolutionary history. However, the very large number of gene death events suggests a second possibility. It is possible that the rates of gene birth and death may be more balanced, i.e, instead of being born once and recurrently lost, a subset of de novo genes may be born, die, and subsequently reborn (*Schmitz et al., 2018*; *Stein et al., 2018*; *Durand et al., 2019*). Thus, some de novo gene evolution may not proceed unidirectionally but rather like a random walk along a continuum. This is consistent with the proto-gene model (*Carvunis et al., 2012*) or the pre-adaptation model (*Wilson et al., 2017*). Under the proto-gene model, the frequent deaths of our utORFs would imply that they are proto-genes and not genes. However, with the exception of the *melanogaster*-specific utORFs, our utORFs are often conserved across species with significant divergence times, which implies that some utORFs may have evolutionary constraints and may, in fact, be functional genes. It is difficult to make strong conclusions here without getting trapped in the philosophical quagmire of the nature of 'gene-ness;' nevertheless, it is possible that no single 'grand unified theory' of de novo gene origination can describe all phenomena.

## Outlook

Together, our results show that evolution of young proteins may progress along different, distinct trajectories in *Drosophila*, as illustrated by the differences between classes described above. Whether similarly distinct properties and trajectories, e.g., with respect to length, age, turnover, and speed of evolution, are apparent in other model species such as yeast or mammals remains to be seen. Of note, *Drosophila* is a taxon of multicellular organisms with a large effective population size, so selective processes are more efficient; mammals – especially primates and *Homo* – are evolutionarily young and have a smaller effective population size, while yeasts are unicellular. If these factors affect general evolutionary properties, such as the selective cost of translation of lowly functional proteins and the probability of fixation by drift, it is possible that they may affect the evolution of de novo proteins. In the case of *Homo*, all these factors may be more favorable to the fixation of new de novo proteins, and the availability of broad and varied omics data is unparalleled. It would therefore be an obvious extension to employ a similar approach to investigate possible utORFs and de novo proteins in humans.

## Methods

### Comprehensive *D. melanogaster* ORFeome

To generate the comprehensive database of potential ORFs in *D. melanogaster*, we translated the repeat-masked genome (UCSC August 2014 BDGP Release 6+ISO1 MT/dm6) and transcriptome (FlyBase, r6.15). We used the repeat-masked genome to control the number of transposable elements and other degenerate candidates. We retained only ORFs of at least eight residues; for ORFs not containing a canonical start codon, only those of at least 20 residues were retained. We deduplicated potential ORF sequences, leaving 4,583,941 unique potential ORFs.

## Discovery of utORFs in published shotgun MS data

To identify evidence of utORFs, we reanalyzed a published proteome of *D. melanogaster* (PRIDE accession numbers PXD005691 and PXD005713; respectively, whole lifecycle and hourly/bihourly embryogenesis stages; *Casas-Vila et al., 2017*) using MaxQuant v. 1.6.1.0 (*Tyanova et al., 2016*). To speed computation, we divided analysis of the overall proteome into four subsets (*Figure 7—source data 3*). As in *Casas-Vila et al., 2017*, we used standard parameters (namely, methionine oxidation and N-terminal acetylation as variable modifications; carbamidomethylation as a fixed modification; tryptic digestion; minimum peptide length 7).

## Two-round MS search

In the two-round MS search, we used two rounds of analysis per data subset to improve total sensitivity while maintaining an acceptable FDR (*Figure 1A*). In the first 'discovery' round, we set the peptide and protein group FDRs in MaxQuant to 0.2, with the protein search sequences set to be the annotated *D. melanogaster* proteome (FlyBase, r6.15), supplemented with the comprehensive potential ORF database. We then identified candidate ORFs from any of the four data subsets (*Figure 7—source data 3*) to develop a filtered subset of potential ORFs with possible MS evidence. These relaxed parameters allow us to identify many more potential ORFs but with a corresponding increase in false-positive identifications. In the second 'verification' round, we set the peptide and protein group FDRs to 0.01, with the protein search sequences set to be the annotated *D. melanogaster* proteome (FlyBase, r6.15), supplemented with only that filtered subset of potential ORFs identified in the first round (around 1000-fold fewer). We subsequently performed additional manual curation of the identified hits to arrive at our final set of utORFs; in brief, we dropped utORFs that were shorter than 14 residues that had detectable sequence homology to annotated proteins via BLASTP, and the shorter of multiple possible utORFs that were identified from the same protein group by MaxQuant.

## Computation of protein secondary structure

To calculate protein structure characteristics, we used DeepCNF (*Wang et al., 2016*) to predict the proportion of helix, sheet, and coiled-coil secondary structure of each gene and to predict the percent of intrinsic structural disorder. For statistical testing, we used a length-matched set of potential ORFs as an empirical null. Specifically, for each utORF, we drew 250 potential ORFs of identical or similar length without replacement, and we calculated the empirical percentile rank of the utORF's metric of interest (percent of coiled-coil and percent of disorder) to yield an empirical p-value estimate. We then used Fisher's method to calculate combined p-values.

## Evaluation of peptide and protein polymorphisms

Evaluating whether a given peptide could have arisen from a polymorphism is equivalent to calculating the number of substitutions from said peptide to the nearest peptide expected from the annotated proteome. Variations of this problem are well studied in computer science, and trie data structures are commonly used to solve them efficiently. Tries, which are also called prefix trees, enable dramatically fewer comparisons than naive brute-force approaches (and thus, substantially faster computation time) at the expense of greater memory usage. We used a Python implementation of tries (https://github.com/jfjlaros/dict-trie; *Laros, 2022*) to build a trie of all tryptic peptides that would be expected from the annotated *D. melanogaster* proteome. For each peptide uniquely identified from a utORF, we calculated Hamming (the number of substitutions without the possibility of indels) and Levenshtein (the combined number of substitutions and indels) distances.

## Genomic analysis of utORFs

To map utORF sequences back to their genomic locations, we used BLAT version 36 (*Hinrichs et al., 2006*) to identify the locations of utORF sequences that were supported in the discovery round. We reformatted the output to a GTF and made certain manual corrections; this file, containing the final locations, is appended as *Figure 1—source data 3*. We excluded those loci whose genomic location could not be determined from genomic analyses. For those loci where multiple genomic locations were possible, we used the first location identified by BLAT.

We subsequently classified the genomic locations of these utORF sequences using custom scripts relying on the Bioconductor libraries in R (*Huber et al., 2015*). Specifically, we classified a location as

sense or antisense if any part of the location overlapped with an annotated gene in the appropriate direction. We used the FlyBase r6.15 annotations. We classified a location as intergenic if and only if it was neither sense nor antisense as defined above.

We computed conservation scores by using the 27-way phastCons and phyloP bigWig tracks provided by the UCSC Genome Browser (*Hinrichs et al., 2006*). For a given genomic location, we calculated the conservation score by averaging the score through the relevant interval.

## Published sequencing data

We used a number of published sequencing datasets to characterize utORFs. To evaluate developmental stage- and tissue-specific transcription levels, we used modENCODE (modENCODE *Roy et al., 2010*) and FlyAtlas2 (*Leader et al., 2018*) data.

## Gene age inference

### MSA and genomic similarity scoring

We used BioPython v. 1.69 (*Cock et al., 2009*) to subset the UCSC 27-way insect MSA to a smaller MSA that corresponded to the location in the *D. melanogaster* genome of each utORF plus the flanking 9 bp on each side. In a pairwise fashion for each species against *D. melanogaster*, we extracted genomic sequences from the MSA, dropped gaps in the sequences, and calculated identity scores.

### Modified Smith-Waterman protein similarity scoring

We calculated protein similarity scores using the Smith-Waterman algorithm with the BLOSUM62 matrix (*Henikoff and Henikoff, 1992*), no gap or extension penalties, and we normalized by the length of the ORF in *D. melanogaster*. To ensure score stability with respect to minor indels, we calculated protein similarity scores in all three reading frames, keeping only the frame that scored best. In doing so, we considered the entirety of the syntenic block irrespective of the formal boundaries of ORFs in the related species; this allows us to capture the locally optimal (within the block) similarity and tolerate recent acquisition of nonsense or frameshift mutations.

### Phylostratigraphy

Using the reference phylogeny corresponding to the UCSC 27-way insect MSA, we inferred approximate gene emergence dates from the pattern of modified Smith-Waterman protein similarity scores. We defined significance as a score greater than 2.5 points. In turn, we estimated the inferred age of a given ORF as the age of the last common ancestor between *D. melanogaster* and the most distant insect species for which a significantly modified Smith-Waterman protein similarity score was calculated.

## Latent class analysis

We performed LCA (*Collins and Lanza, 2009*) using the *poLCA* R package (*Linzer and Lewis, 2011*). Specifically, we used 100 iterations of a latent class model with no covariates and with factors as follows: inferred gene age; monophyly; genomic location (classified as solely intergenic, antisense, sense, or any combination); phastCons score (binned as [0, 0.2], [0.2, 0.8], and [0.8, 1.0]); length (binned as [0, 20], [20, 50], and [50, infinity]); transcription (binned as TPM >0.1 or ≤0.1 from maximum per-sample TPM); tissue specificity (binned as *tau* >0.8 or ≤0.8 from FlyAtlas tissues); developmental specificity (binned as *tau* >0.8 or ≤0.8 from modENCODE developmental stages). We excluded loci with missing values from LCA. We chose the number of classes (five) based on a combination of minimizing the BIC and AIC (*Supplementary file 1D and E*) as well as model interpretation similar to recommended procedures (*Collins and Lanza, 2009*). For the analysis of all utORFs, models with six classes did not converge on a single maximum-likelihood solution, suggesting potential under-identification, and were thus not considered.

We subsequently used the resulting five-class latent class model to predict the posterior class membership probabilities for each classified candidate locus. We used the modal probability to assign class membership. We also performed a parallel analysis of utORFs with canonical start codons, which led to similar conclusions.

## Validation of utORFs

### Retention time validation

We used AutoRT (*Wen et al., 2020*) to predict retention times of identified peptides from both annotated FlyBase proteins and utORFs. To train the model, we conducted transfer learning per each source run using the published AutoRT model (originally trained on PXD006109) with 40 epochs on either an NVIDIA K80 GPU or an NVIDIA V100 GPU; the training set consisted of only those annotated FlyBase peptides with posterior error probabilities of 0.01 or smaller as estimated by MaxQuant. We then used the per-run trained models with default parameters to predict the retention times of the peptides resulting from either the annotated FlyBase proteins or utORFs.

### Empirical validation

We collected protein samples from *D. melanogaster* 0–24 hr embryos (RAL517 strain) for validation of utORFs via MS. In brief, we homogenized dechorionated embryos with a mortar and pestle in ice-cold lysis buffer (50 mM Tris HCl, pH 7.5; 137 mM NaCl; 0.25% NP-40; Roche cOmplete protease inhibitors) before centrifugation at 18,000 rpm at 4°C and filtration through a 0.45 micron PVDF membrane.

Purified protein was acetone-precipitated, reduced, and alkylated before digestion with LysC and trypsin, acidified, and solid-phase extraction. The resulting peptides were analyzed by LC-MS/MS (120 min gradient, 25 cm EasySprayer column in high res./high mass accuracy mode) on a ThermoFisher Lumos instrument. In addition, the embryo sample was analyzed on the same instrument by PRM-MS, targeting a set of 16 tryptic peptides. We manually chose these targeted peptides based on the suitability of the peptide for targeting (i.e. few/no missed cleavages or modifications) the expression of the utORF in embryos and the strength of the prior detection from the Casas-Vila datasets.

We analyzed the shotgun data using ProteomeDiscoverer 1.4 and searched via Mascot against the Uniprot *D. melanogaster* database concatenated with common contaminants and our set of utORFs with a 1% peptide FDR. We analyzed the targeted data using SkylineMS (v. 21.2.0.369 [2efacf038]). We deposited the raw data in PRIDE under accession PXD032197.

We additionally reanalyzed published ribosome profiling data downloaded from the NCBI Sequence Read Archive (SRA) project PRJNA306373 (*Zhang et al., 2018*) and project SRP254283 (*Patraquim et al., 2020*). For the Zhang dataset, we downloaded raw reads from the above SRA accessions, aligned them to the genome using hisat2 v.2.2.1 (*Kim et al., 2019*), and calculated TPM using stringtie v.1.3.4d (*Pertea et al., 2016*). We handled the Patraquim dataset similarly, except that we additionally trimmed the raw reads using trimmomatic 0.38 (*Bolger et al., 2014*) and filtered out reads aligning to annotated RNAs using bowtie2 v. 2.3.5 (*Langmead and Salzberg, 2012*) before alignment. For each utORF, we calculated the maximum TPM observed across either dataset and called the utORF as supported if the TPM was greater than 0.2.

To identify utORFs with statistically significant framing of ribosome-profiling reads, we used a binomial test as in *Patraquim et al., 2020*. We aligned the trimmed, filtered reads to the set of sequences corresponding to our utORFs, plus an additional 30 bp up- and downstream, using bowtie v. 1.3.1 (*Langmead et al., 2009*) with options – tryhard – suppress 1, 6, 7, and 8. We analyzed the resulting aligned footprints using the R package riboSeqR v 1.20.0. For each utORF with at least two unique footprints, we calculated the binomial probability of observing the number of footprints in the utORF's frame vs. the other two frames at an uncorrected alpha of 0.05. We pooled footprints across replicates and across read lengths for a given utORF in each sample.

## Acknowledgements

We thank members of the Zhao laboratory for their helpful discussions during the work. We thank Junhui Peng for the help with secondary structure prediction. We are grateful to Henrik Molina and the Proteomics Resource Center at The Rockefeller University for the help with mass spectrometry data.

## Additional information

### Funding

| Funder | Grant reference number | Author |
| --- | --- | --- |
| National Institute of General Medical Sciences | R35GM133780 | Li Zhao |
| National Institute of General Medical Sciences | T32GM007739 | Eric B Zheng |
| Robertson Foundation | | Li Zhao |
| Rita Allen Foundation | Rita Allen Foundation Scholar | Li Zhao |
| Vallee Foundation | Vallee Scholar | Li Zhao |
| Monique Weill-Caulier Trust | | Li Zhao |
| Alfred P. Sloan Foundation | Alfred P. Sloan Research Fellowship | Li Zhao |

The funders had no role in study design, data collection and interpretation, or the decision to submit the work for publication.

### Author contributions

Eric B Zheng, Conceptualization, Data curation, Software, Formal analysis, Validation, Investigation, Visualization, Methodology, Writing - original draft; Li Zhao, Conceptualization, Resources, Funding acquisition, Methodology, Writing - original draft

### Author ORCIDs

Eric B Zheng (iD) http://orcid.org/0000-0002-6302-3635
Li Zhao (iD) http://orcid.org/0000-0001-6776-1996

### Decision letter and Author response

Decision letter https://doi.org/10.7554/eLife.78772.sa1
Author response https://doi.org/10.7554/eLife.78772.sa2

## Additional files

### Supplementary files
- MDAR checklist
- Supplementary file 1. Supplementary tables 1A–1E.

### Data availability

Raw MS data are deposited in PRIDE under accession number PXD032197. Relevant scripts and intermediate files can be found in our Github repository (https://github.com/LiZhaoLab/utORF_mass_spec, copy archived at swh:1:rev:a1f7c82dc23161d0df31fa4d0e4cd38b0c59fb48).

The following dataset was generated:

| Author(s) | Year | Dataset title | Dataset URL | Database and Identifier |
| --- | --- | --- | --- | --- |
| Zheng EB, Zhao L | 2022 | Verification of utORFs in *Drosophila melanogaster* | http://proteomecentral.proteomexchange.org/cgi/GetDataset?ID=PXD032197 | ProteomeXchange, PXD032197 |

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
