## [Editor Report]

By integrating in silico predictions and mass-spectrometry, this manuscript tackles the problem of annotating the currently nameless stretches of genomic sequence that actually code for proteins. The hundreds of fruit fly protein-coding genes described here offer new inroads for studying some of the very youngest functional elements in genomes, particularly those that have recently emerged from non-coding DNA sequences. This report will be of interest to functional and evolutionary genomics communities.

---

## [Decision Letter]

**Decision letter after peer review:**

Thank you for submitting your article "Systematic identification of unannotated ORFs in *Drosophila* reveals evolutionary heterogeneity" for consideration by *eLife*. Your article has been reviewed by 3 peer reviewers, and the evaluation has been overseen by a Reviewing Editor and Molly Przeworski as the Senior Editor. The reviewers have opted to remain anonymous.

Essential Revisions:

1) The question of the biological significance of genes identified here should be clearly articulated in a single, streamlined section. The authors should offer a list of genes that, based on their criteria (e.g., age, lack of presence/absence polymorphism in population datasets, retention time compared to annotated ORFs), are likely to be biologically significant. The authors should offer the reader a clear path to select genes for downstream functional analysis.

2) Related: the section on independent validation requires modification. This section leaves the reader wondering if only a tiny fraction of the genes described here are biologically significant. If expression heterogeneity accounts for the low rate of validation, then only utORFs expressed at the same developmental stage of the MS experiment should have been selected. Otherwise, this section should be removed.

3) The study's significance is not sufficiently clear from the text. Genome-wide descriptions of utORFs have been conducted in many species, including *Drosophila* (PMID: 25144939; PMID: 32471506). The impact of the present study can only be inferred by comparisons to other studies using other methods. A quantitative comparison of unannotated ORFs between this study and other published studies in *Drosophila*, e.g. PMID: 25144939; PMID: 32471506 should be performed. The inference(s) from these comparisons should be used to highlight the importance of the present study, assuming that a new group of utORFs – that are likely biologically significant-have been identified. Large overlap between the genes described here and these previous studies would diminish the impact of the authors' MS-approach.

4) The choice to proceed with the 993 utORFs needs more justification. Please cite literature on the functional significance of genes that employ non-canonical start codons, which such codons, and whether such codons are overrepresented in the current dataset of non-ATG start codon proteins. At present, it's not clear why the authors didn't proceed with only the 300+ subset of utORFs with canonical start codons to ensure that inferences are biologically relevant.

5) The abstract and introduction open with the problem of de novo gene evolution. However, this topic represented only a small fraction of the Results section. This mismatch of framing and data presented should be resolved.

6) The title of the study should be modified. A discovery of "heterogeneity" is not of sufficient importance/impact to warrant an appearance in the title. Please revise.

7) The Discussion section is too long and takes away from the major points of the manuscript. Please condense and/or shunt much of the text to a supplement. For example, the final section on methodological considerations should be revised heavily/removed.

8) Claims made from the LCA analysis should be retested on the entire dataset without regard to the LCA bins. For example, the age-length comparison, evolutionary rate v age, etc.

Additional Comments and Questions:

9) Other mechanisms besides de novo birth may contribute to lineage-specific genes that have no homology in related genomes. These include the horizontal transfer from rapidly evolving donor like bacteria or virus (an analysis of experimental *E. coli* evolution data, the 2022 *Drosophila* conference); the pseudogenization and loss of homologous genes in related species (Ma et al., 2021. Dev Cell 56: 1-16). (see also Zhang et al., (2019. Nature Ecol Evol. 3: 679-690). Please include these alternatives as potential mechanisms.

10) Please clarify the developmental stages used in each of the datasets.

11) Line 104: "…utORF sequences have qualitatively similar amino acid composition (Figure 1D). Why not replace this not so scientific word "qualitatively" when it is easy to do a statistical test to test significance?

12) TMP >0.2 in the methods section is not sufficient for the expression of unannotated ORFs. A strong 3-nt periodicity should be considered as acceptable evidence for active translation of identified ORFs.

13) Lines 249-250, these are interesting observations but may not be that surprising, given the published observations of new genes in *Drosophila* brain and their behavioral impacts (e.g. Chen et al., 2012. Cell Report 1(2): 118-132; Dai et al., 2007. PNAS 105(21).7478-7483).

14) Where are the data found that support the claim: "gene age inferences are not dramatically affected by choice of significance threshold." (l.179, p.6).

15) Please clarify the following methodological points:

For optimal alignments, does the search include only ORFs in each orthologous region?

Why not require two outgroups to infer de novo origin (as they cited is important?)

16) Please provide data supporting the claims:

a. fastest evolving utORFs are not the youngest evolutionarily.

b. more ORFs expressed in the brain than the testis. This point also suffers by a lack of appropriate statistical comparisons.

17) Previous work such as that by Vakirlis et al., 2020, *eLife* should be cited in the context of sequence divergence erasing detectable similarity. Given these recent advances, is this issue still of great importance?

---

## [Author Response]

Essential Revisions:1) The question of the biological significance of genes identified here should be clearly articulated in a single, streamlined section. The authors should offer a list of genes that, based on their criteria (e.g., age, lack of presence/absence polymorphism in population datasets, retention time compared to annotated ORFs), are likely to be biologically significant. The authors should offer the reader a clear path to select genes for downstream functional analysis.

Thank you for the suggestions. We have added new text at the end of the independent validation section of the results (see Lines 374-384). We use the combination of high transcription from published RNAseq atlases, either MS re-validation or framed ribosome profiling re-validation, and conservation metrics to define our priority for functional studies. In Figure 7 – Source Data File 2, we report these results, including a column that indicates priority for future work.

We have also revised the independent validation section of the results as well as the discussion to clarify our views on the overall biological significance of utORFs (See responses to items 2, 3, 4, 5, 7, 9, 12, 15, and 17 for individual details.) In brief, we are confident in our identified utORFs, but we believe that when planning downstream functional analysis, it is best to target those with a combination of both strong experimental support and interesting conservation (both high and low conservation) in a tissue/developmental stage that robust experimental assays such as a fitness assay can be reasonably defined.

2) Related: the section on independent validation requires modification. This section leaves the reader wondering if only a tiny fraction of the genes described here are biologically significant. If expression heterogeneity accounts for the low rate of validation, then only utORFs expressed at the same developmental stage of the MS experiment should have been selected. Otherwise, this section should be removed.

We apologize for the confusion about the MS experiments: we did indeed target only utORFs expressed in the embryo, and we used a sample of embryos of mixed ages from 0-24 hours for both the targeted and the shotgun MS experiment. We have revised the text to directly state this without referencing the methods section (Lines 353-355; 357-359). We also realized that the word “heterogeneity” can be confusing, so we have eliminated it from the manuscript.

We also have added a new framing re-analysis of the Patraquim data (see item 3) that further supports our utORFs. Given that MS methods in particular have notoriously low coverage even for canonical genes in humans, which is the best-studied organism for MS (Baker et al., Nat. Comm. 2016; PMID: 20411723), we believe that our validation rate is a conservative lower bound for the biological relevance of our utORFs. We have additionally revised our discussion to clarify this (Lines 428-437).

3) The study's significance is not sufficiently clear from the text. Genome-wide descriptions of utORFs have been conducted in many species, including *Drosophila* (PMID: 25144939; PMID: 32471506). The impact of the present study can only be inferred by comparisons to other studies using other methods. A quantitative comparison of unannotated ORFs between this study and other published studies in *Drosophila*, e.g. PMID: 25144939; PMID: 32471506 should be performed. The inference(s) from these comparisons should be used to highlight the importance of the present study, assuming that a new group of utORFs – that are likely biologically significant-have been identified. Large overlap between the genes described here and these previous studies would diminish the impact of the authors' MS-approach.

Thank you for the suggestion. Our hits actually have no overlaps with those reported in the supplementary information from Patraquim et al., 2020 (PMID: 32471506, which contains the data from PMID: 25144939). However, we note that they began with FlyBase-annotated CDSs and focused on the longest ORF per CDS with a canonical start codon. Our work begins with a six-frame translation of the genome, so it contains both the ORFs considered by Patraquim et al., and a far greater number of additional putative ORFs, most of which were not included in the Patraquim et al., analytical pipeline. We further improved our analysis by using the raw data from Patraquim et al., and implementing their binomial probability test for statistically significant framing (Lines 369-373; methods in Lines 639-646), which yields support for another 17 utORFs of 81 with footprints in the embryo. Note that Patraquim et al., amplified sequences from polysomes, so there will not be a hit if the transcript of a utORF is occupied by monosomes; since many of our utORFs are very short, it is possible they are preferentially translated by monosomes instead of polysomes (see also Heyer & Moore 2016, PMID: 26871635, or in certain tissue, see Biever et al., PMID: 32001627). We believe that this both supports our proposed MS-first approach and highlights its potential for uncovering otherwise-missed genic loci.

The reviewers very accurately pointed out that the data from the two referenced papers about ribosome profiling contain periodicity (see also item 12). However, several other ribosome profiling datasets in *Drosophila* have a greater depth in terms of developmental stages and amount of data, but their data does not have periodicity information because they used a different enzyme. While we agree with reviewers that using ribosome profiling data with periodicity is ideal, we also believe that the other high-quality data without periodicity information remain very useful for this work. We thus kept those analyses while adding the analyses suggested by the reviewers to use periodicity data. As suggested, we have also edited the manuscript to highlight the significance of our work.

4) The choice to proceed with the 993 utORFs needs more justification. Please cite literature on the functional significance of genes that employ non-canonical start codons, which such codons, and whether such codons are overrepresented in the current dataset of non-ATG start codon proteins. At present, it's not clear why the authors didn't proceed with only the 300+ subset of utORFs with canonical start codons to ensure that inferences are biologically relevant.

We have added literature about non-canonical start codons. One main reason for the inclusion of utORFs with non-canonical start codons is that hits with non-canonical start codons are understudied, which has obvious relevance in the context of unrevealed protein evolutionary innovation. In the revised version, we have added a parallel analysis to the latent class analysis (Figures 4 and 5) that uses exclusively utORFs with canonical start codons. The results are qualitatively similar, with the major exception that short utORFs (which are often in the fast-evolving class) tend to not have canonical start codons. As a result, certain trends that we report with regards to that class are affected by considering only utORFs with canonical start codons. We report all of these in the revised LCA Results section (Figure 4—figure supplement 2, Figure 4—figure supplement 3, Figure 5—figure supplement 2; and Lines 290-309).

5) The abstract and introduction open with the problem of de novo gene evolution. However, this topic represented only a small fraction of the Results section. This mismatch of framing and data presented should be resolved.

Thank you for this suggestion. We have rewritten the introduction. We now use a more historical opening to focus broadly on lineage-specific genes and evolutionarily young genes and frame de novo genes in that context. We also explicitly acknowledge other mechanisms of young gene evolution. We believe the new opening is much stronger.

6) The title of the study should be modified. A discovery of "heterogeneity" is not of sufficient importance/impact to warrant an appearance in the title. Please revise.

We have changed the title to “Protein evidence of unannotated ORFs in *Drosophila* reveals unappreciated diversity in the evolution of young proteins.”

7) The Discussion section is too long and takes away from the major points of the manuscript. Please condense and/or shunt much of the text to a supplement. For example, the final section on methodological considerations should be revised heavily/removed.

We have substantially revised the Discussion section. The aforementioned final section has been removed, and the most important elements have been consolidated with relevant sections. We have also used some of the space to expand on other aspects that were suggested by the reviewers during the review.

8) Claims made from the LCA analysis should be retested on the entire dataset without regard to the LCA bins. For example, the age-length comparison, evolutionary rate v age, etc.

Based on this comment and the specific comment from Reviewer 3, we believe that this concern is rooted in our failure to clearly explain the motivation for Figure 5. We agree that binning continuous variables could distort the underlying data. However, in Figure 5 and in the associated statistical tests, we conduct our comparisons on the original continuous variables, not on the binned data used for LCA. We have inserted new language at the transition paragraph (Lines 243-247) to explain this.

Moreover, the trends we describe are not observable by simple pairwise comparisons. Author response image 1 illustrates the length-vs-age comparison, phyloP-vs-age comparison, and phastCons-vs-age comparison, colored by LCA-inferred class for all utORFs. This was the motivation for our adoption of Latent Class Analysis.

**Author response image 1. sa2fig1:** 

Additional Comments and Questions:9) Other mechanisms besides de novo birth may contribute to lineage-specific genes that have no homology in related genomes. These include the horizontal transfer from rapidly evolving donor like bacteria or virus (an analysis of experimental *E. coli* evolution data, the 2022 *Drosophila* conference); the pseudogenization and loss of homologous genes in related species (Ma et al., 2021. Dev Cell 56: 1-16). (see also Zhang et al., (2019. Nature Ecol Evol. 3: 679-690). Please include these alternatives as potential mechanisms.

We agree that a discussion of other mechanisms that could yield apparent lineage-specific genes would be valuable, and we thank the reviewer for bringing certain fascinating examples to our attention. We have inserted a new paragraph in the discussion discussing this (Lines 414-423). Work by others, as highlighted by another reviewer (see item 17), and our own implies that the utORFs we describe here are unlikely to be the result of substantial divergence.

10) Please clarify the developmental stages used in each of the datasets.

We have clarified in the manuscript the exact developmental stages used in both the Casas-Vila MS data (Lines 108-111) as well as our targeted MS (Lines 353-366; 357-359). We have also inserted a column in Figure 1 – Source Data File 4 explaining the developmental stage in each source MS file.

11) Line 104: “…utORF sequences have qualitatively similar amino acid composition (Figure 1D). Why not replace this not so scientific word “qualitatively” when it is easy to do a statistical test to test significance?

Thank you for the suggestion. We have added statistical testing, using Spearman rank-order correlations as a proxy for similarity in the amino acid frequencies (Lines 133-134; 143-145), and we have removed the word “qualitatively”.

12) TMP >0.2 in the methods section is not sufficient for the expression of unannotated ORFs. A strong 3-nt periodicity should be considered as acceptable evidence for active translation of identified ORFs.

When we examined ribosome profiling data for random intergenic sequences, we found that the average TPM was much lower than 0.2 and was very near to 0, indicating that ribosome profiling does not have many artifacts. It is possible that non-coding sequences may also be protected by ribosomes, in which case ribosome profiling alone would not be sufficient to prove the translation of an ORF. However, since we are studying sequences that have evidence of translation from MS, we believe that the issue here is less critical. Another practical reason is that several high-quality ribosome profiling datasets from *Drosophila* do not have periodicity, but it would be a pity to ignore them. We do strongly agree with the reviewers that periodicity is important, and as the reviewers mentioned, two papers from one group contain data from a few samples. In addition, we enhanced our analysis of the Patraquim data by implementing their binomial probability test for statistically significant framing (Lines 369-373; methods in Lines 639-646), which yields support for another 17 utORFs in the embryo. We hope this helps reassure the reviewer.

13) Lines 249-250, these are interesting observations but may not be that surprising, given the published observations of new genes in *Drosophila* brain and their behavioral impacts (e.g. Chen et al., 2012. Cell Report 1(2): 118-132; Dai et al., 2007. PNAS 105(21).7478-7483).

We agree that we may have overstated the level of surprise and the relevant context. We have adjusted our language (Lines 286-289), and we have also revised our discussion to clarify the distinction between all types of evolutionarily young genes.

14) Where are the data found that support the claim: “gene age inferences are not dramatically affected by choice of significance threshold.” (l.179, p.6).

Thank you for the reminder. We have added the necessary supplemental figure (Lines 210-211; Figure 3—figure supplement 1).

15) Please clarify the following methodological points:For optimal alignments, does the search include only ORFs in each orthologous region?Why not require two outgroups to infer de novo origin (as they cited is important?)

For optimal alignments, our search considers the entirety of the syntenic block in all three parallel frames, with gaps dropped. We did this since enforcing ORF boundaries would dramatically reduce the search space (and thus reduce the computational advantage gained by using the syntenic block to focus an optimal algorithm). Moreover, comparing only ORFs would be brittle to recent and/or frequent acquisition of nonsense or frameshift mutations; our results regarding turnover suggest that this is a plausible concern. We have revised the methods section to clarify (Lines 577-580).

We have increased our stringency for inferring de novo origin; we now require two supporting outgroups that cannot be from the same taxon (that is, they cannot be sister species of each other). Our results remain highly similar (see the new Figure 6).

16) Please provide data supporting the claims:a. fastest evolving utORFs are not the youngest evolutionarily.b. more ORFs expressed in the brain than the testis. This point also suffers by a lack of appropriate statistical comparisons.

We have clarified the language on both points, and we have added a new statistical test (Lines 281-285).

17) Previous work such as that by Vakirlis et al., 2020, eLife should be cited in the context of sequence divergence erasing detectable similarity. Given these recent advances, is this issue still of great importance?

We have added a reference to Vakirlis et al., 2020, *eLife* in the manuscript as part of our revised Discussion section examining mechanisms that yield lineage-specific genes (Lines 414-423). Vakirlis et al., 2020 was an important advance in our understanding of the interplay between sequence divergence, detectable similarity, and inferences of mechanisms of gene evolution. While we agree with the work by Vakirlis *et al.,* some researchers remain concerned about whether other mechanisms beyond sequence divergence can yield apparent lineage-specific genes. In the context of our work, synteny is even more important since many utORFs are short, which might be more challenging technically to study than the genes studied by Vakirlis et al., 2020.